# DISENTANGLED ACTIVATIONS IN DEEP NETWORKS

## ABSTRACT

Deep neural networks have been tremendously successful in a number of tasks. One of the main reasons for this is their capability to automatically learn representations of data in levels of abstraction, increasingly disentangling the data as the internal transformations are applied. In this paper we propose a novel regularization method that penalize covariance between dimensions of the hidden layers in a network, something that benefits the disentanglement. This makes the network learn nonlinear representations that are linearly uncorrelated, yet allows the model to obtain good results on a number of tasks, as demonstrated by our experimental evaluation. The proposed technique can be used to find the dimensionality of the underlying data, because it effectively disables dimensions that aren't needed. Our approach is simple and computationally cheap, as it can be applied as a regularizer to any gradient-based learning model.

## 1 INTRODUCTION

A good data representation should ultimately uncover underlying factors in the raw data while being useful for a model to solve some task. Deep neural networks learn representations that are increasingly abstract in deeper layers, disentangling the causes of variation in the underlying data (Bengio et al., 2013). Formal definitions of disentanglement are lacking, although Ver Steeg & Galstyan (2015); Achille & Soatto (2017) both use the total correlation as a measure of disentanglement. Inspired by this, we consider a simpler objective: a representation disentangles the data well when its components do not correlate, and we explore the effects of penalizing this linear dependence between different dimensions in the representation. Ensuring independence in the representation space results in a distribution that is factorizable and thus easy to model (Kingma & Welling, 2014; Rezende et al., 2014).

We propose a novel regularization scheme that penalizes the cross-correlation between the dimensions of the learned representations, and helps artificial neural networks learn disentangled representations. The approach is very versatile and can be applied to any gradient-based machine learning model that learns its own distributed vector representations. A large body of literature have been published about techniques for learning non-linear independent representations (Lappalainen & Honkela, 2000; Honkela & Valpola, 2005; Dinh et al., 2015), but in comparison our approach is simpler, and does not impose restrictions on the model used. The proposed technique penalizes representations with correlated activations. It strongly encourages the model to find the dimensionality of the data, and thus to disable superfluous dimensions in the resulting representations. The experimental evaluation on synthetic data verifies this: the model is able to learn all useful dimensions in the data, and after convergence, these are the only ones that are active. This can be of great utility when pruning a network, or to decide when a network needs a larger capacity. The disabling of activations in the internal representation can be viewed as (and used for) dimensionality reduction. The proposed approach allows for interpretability of the activations computed in the model, such as isolating specific underlying factors. The solution is computationally cheap, and can be applied without modification to many gradient-based machine learning models that learns distributed representations.

Moreover, we present an extensive experimental evaluation on a range of tasks on different data modalities, which shows that the proposed approach disentangles the data well; we do get uncorrelated components in the resulting internal representations, while retaining the performance of the models on their respective task.

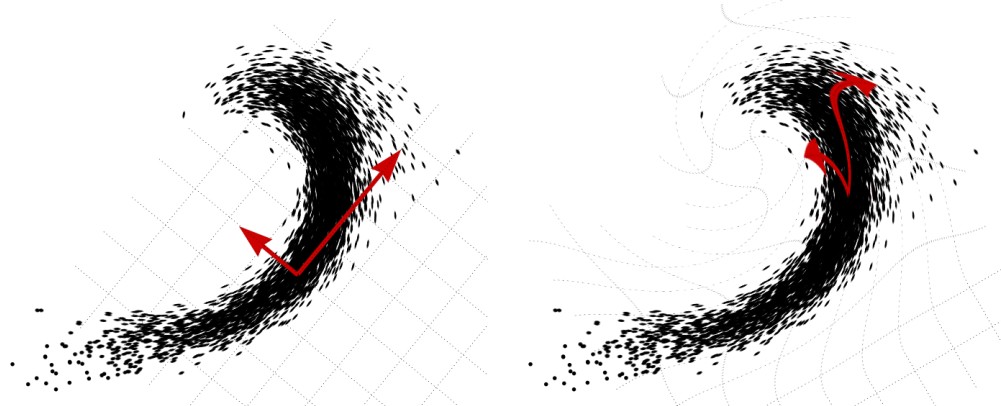

Figure 1: When data is distributed along non-linear manifolds, a linear model cannot describe the data well (left). However, with a non-linear model (right), it is possible to capture the variations of the data in a more reasonable way and unfold it into a compact orthogonal representation space.

The main contributions of this work include: $L_\Sigma$ regularization, a novel approach penalizing the covariance between dimensions in a representation (see Section 2). The regularizer encourages a model to use the minimal number of dimensions needed in the representation. The approach is computationally cheap and can be applied without any restrictions on the model. The experimental evaluation shows how different models can benefit from using $L_\Sigma$ regularization. From autoencoders on synthetic data to deep convolutional autoencoders trained on CIFAR-10, we show that $L_\Sigma$ helps us learn uncorrelated and disentangled representations (see Section 3).

## 2 DISENTANGLEMENT THROUGH PENALIZING CROSS-CORRELATIONS

We present a novel regularizer based on the covariance of the activations in a neural network layer over a batch of examples. The aim of the regularizer is to penalize the covariance between dimensions in the layer to decrease linear correlation.

### 2.1 DEFINITION

The covariance regularization term ($L_\Sigma$) for a layer, henceforth referred to as the coding layer, is computed as

$$L_\Sigma = \frac{1}{p^2}||\mathcal{C}||_1 \tag{1}$$

where $p$ is the dimensionality of the coding layer,

$$||\mathcal{C}||_1 = \sum_{i,j=1}^{N} |\mathcal{C}_{ij}|, \tag{2}$$

is the element wise L1 matrix norm of $\mathcal{C}$, and $\mathcal{C} \in \mathcal{R}^{p \times p}$ is the sample covariance of the activations in the coding layer over $N$ examples

$$\mathcal{C} = \frac{1}{N-1} \sum_{i=1}^{N} (\mathbf{H} - \mathbf{1}_N \bar{\mathbf{h}})^T (\mathbf{H} - \mathbf{1}_N \bar{\mathbf{h}}).$$

Further, $\mathbf{H} = [\mathbf{h}_1; ...; \mathbf{h}_N]$ is a matrix of all activations in the batch, $\mathbf{1}_N$ is an $N$-dimensional column vector of ones, and $\bar{\mathbf{h}}$ is the mean activation.

### 2.2 USAGE

As $L_\Sigma$ has the structure of a regularizer, it can be applied to most gradient based models without changing the underlying architecture. In particular, $L_\Sigma$ is simply computed based on select layers and added to the error function, e.g. $Loss = Error + \lambda L_\Sigma$

## 3 EXPERIMENTS

This section describes the experimental evaluation performed using $L_\Sigma$ regularization on different models in various settings, from simple multi-layer perceptron-based models using synthetic data (see Section 3.2 and 3.3) to convolutional autoencoders on real data (see Section 3.4). However, before describing the experiments in detail we define the metrics that will be used to quantify the results.

### 3.1 EVALUATION METRICS

A number of different metrics are employed in the experiments to measure different aspects of the results.

**Mean Absolute Pearson Correlation (MAPC)**   Pearson correlation report the normalized linear correlation between variables $\in [-1, 1]$ where 0 indicates no correlation. To get the total linear correlation between all dimensions in the coding layer the absolute value of each contribution is averaged.

$$\mathbf{MAPC} = \frac{2}{(p^2 - p)} \sum_{i<j}^{p} \frac{|\mathcal{C}_{ij}|}{\sqrt{\mathcal{C}_{ii}}\sqrt{\mathcal{C}_{jj}}}$$

**Covariance/Variance Ratio (CVR)**   Though mean absolute Pearson correlation measure the quantity we are interested in it becomes ill defined when the variance of one (or both) of the variables approaches zero. To avoid this problem we define a related measure where all variances are summed for each term. Hence, as long as some dimension has activity the measure remains well defined. More precise, the CVR score is computed as:

$$\mathbf{CVR} = \frac{1}{p^2} \frac{||\mathcal{C}||_1}{\mathbf{tr}(\mathcal{C})}$$

where $||\mathcal{C}||_1$ is defined as in Equation 2. The intuition behind CVR is simply to measure the fraction of all information that is captured in a linear uncorrelated fashion within the coding layer.

**Top d-dimension Variance/total variance (TdV)**   TdV measure to what degree the total variance is captured inside the variance of the top $d$ dimensions. When $d$ is equal to the actual dimension of the underlying data this measure is bounded in [0,1].

**Utilized Dimensions (UD)**   UD is the number of dimensions that needs to be kept to retain a set percentage, e.g. 90% in the case of $UD_{90\%}$, of the total variance. This measure has the advantage that the dimension of the underlying data does not need to be known a priori.

### 3.2 DIMENSIONALITY REDUCTION

The purpose of this experiment is to investigate if it is possible to disentangle independent data that has been projected to a higher dimension using a random projection, i.e. we would like to find the principal components of the original data.

The model we employ in this experiment is an auto encoder consisting of a linear $p = 10$ dimensional coding layer and a linear outputlayer. The model is trained using the proposed covariance regularization $L_\Sigma$ on the coding layer.

The data is generated by sampling a $d = 4$ dimensional vector of independent features $z \sim N(0, \Sigma)$, where $\Sigma \in \mathcal{R}^{d \times d}$ is constrained to be non-degenerate and diagonal. However, before the data is fed to the autoencoder it is pushed through a random linear transformation $x = \Omega z$. The goal of the model is to reconstruct properties of $z$ in the coding layer while only having access to $x$.

The model is trained on 10000 iid random samples for 10000 epochs. 9 experiments were performed with different values for the regularization constant $\lambda$. The first point on each curve (in Figure 2 and 3) is $\lambda = 0$, i.e. no regularization, followed by 8 points logarithmically spaced between 0.001 and

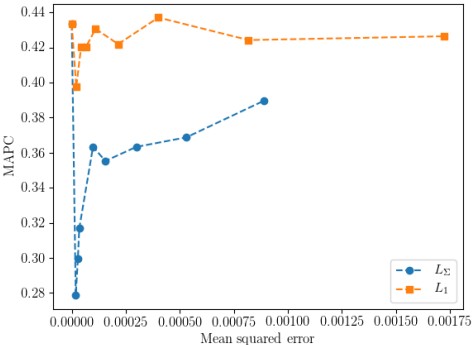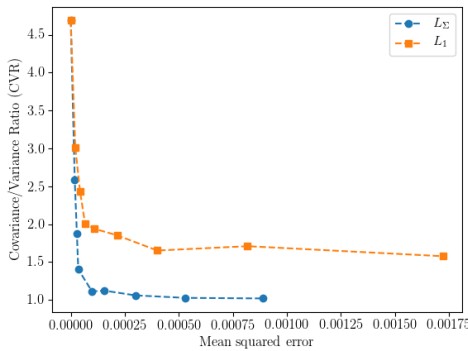

Figure 2: In this figure we compare the amount of residual linear correlation after training the model with $L_\Sigma$ and $L_1$ regularization respectively, measured in MAPC (left) and CVR (right). The first point on each curve corresponds to $\lambda = 0$, i.e. no regularization, followed by 8 points logarithmically spaced between 0.001 and 1. All scores are averaged over 10 experiments using a different random projection ($\Omega$).

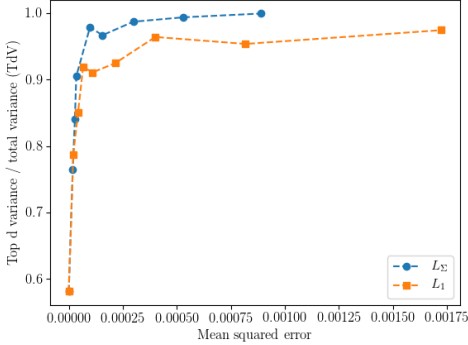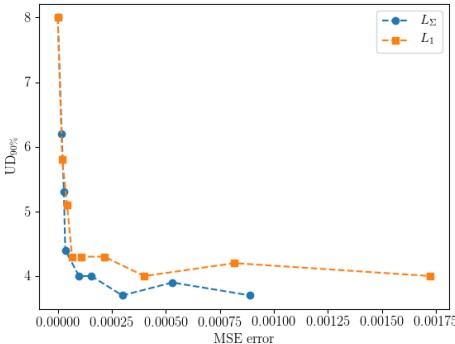

Figure 3: The resulting dimensionality the coding layer after training the model with $L_\Sigma$ and $L_1$ regularization respectively, measured in TdV (left) and $UD_{90\%}$ (right). The first point on each curve corresponds to $\lambda = 0$, i.e. no regularization, followed by 8 points logarithmically spaced between 0.001 and 1.All scores are averaged over 10 experiments using a different random projection ($\Omega$).

1. Each experiment is repeated 10 times using a different random projection $\Omega$ and the average is reported.

The result of the experiment is reported using all four metrics defined in Section 3.1. The result in terms of MAPC and CVR is reported in Figure 2. The first thing to notice is that $L_\Sigma$ consistently lead to lower correlation while incurring less MSE penalty compared to $L_1$. Further, looking at the MAPC it is interesting to notice that it is optimal for a very small values of $L_\Sigma$. This is because higher amounts of $L_\Sigma$ leads to lowering of the dimensionality of the data, see Figure 3, which in turn yields unpredictable Pearson correlation scores between these inactivated neurons. However, this effect is compensated for in CVR for which $L_\Sigma$ quickly converges towards the optimal value of one, which in turn indicates no presence of linear correlation.

Turning the attention to dimensionality reduction, Figure 3 shows that $L_\Sigma$ consistently outperform $L_1$. Further, looking closer at the TdV score, $L_\Sigma$ is able to compress the data almost perfectly, i.e. TdV=1, at a very small MSE cost while $L_1$ struggle even when accepting a much higher MSE cost. Further, the $UD_{90\%}$ scores again show that $L_\Sigma$ achieves a higher compression at lower MSE cost. In this instance the underlying data was of 4 dimensions which $L_\Sigma$ quickly achieves. At higher amounts of $L_\Sigma$ the dimensionality even locationally fall to 3, however, this is because the threshold is set to 90%.

### 3.3 DEEP NETWORK OF UNCORRELATED FEATURES

In Section 3.2 we showed that we can learn a minimal orthogonal representation of data that is generated to ensure that each dimension is independent. However, in reality it is not always possible to encode the necessary information, to solve the problem at hand, in an uncorrelated coding layer, e.g. the data illustrated in Figure 1 would first need a non linear transform before the coding layer. However, using a deep network it should be possible to learn such a nonlinear transformation that enables uncorrelated features in higher layers. To test this in practice on a problem that has this property but still is small enough to easily understand we turn to the XOR problem.

It is well known that the XOR problem can be solved by a neural network of one hidden layer consisting of a minimum of two units. However, instead of providing this minimal structure we would like the network to discover it by itself during training. Hence, the model used is intentionally over-specified consisting of two hidden layers of four logistic units each followed by a one dimensional logistic output layer.

The model was trained on XOR examples, e.g. [1,0]=1, in a random order until convergence with $L_\Sigma$ applied to both hidden layers and added to the cost function after scaling it with $\lambda = 0.2$.

As can be seen in Figure 4 the model was able to learn the optimal structure of exactly 2 dimensions in the first layer and one dimension in the second. Further, as expected, the first layer do encode a negative covariance between the two active units while the second layer is completely free from covariance. Note that, even though the second hidden layer is not the output of the model it does encode the result in that one active neuron. For comparison, see Figure 5 for the same model trained without $L_\Sigma$.

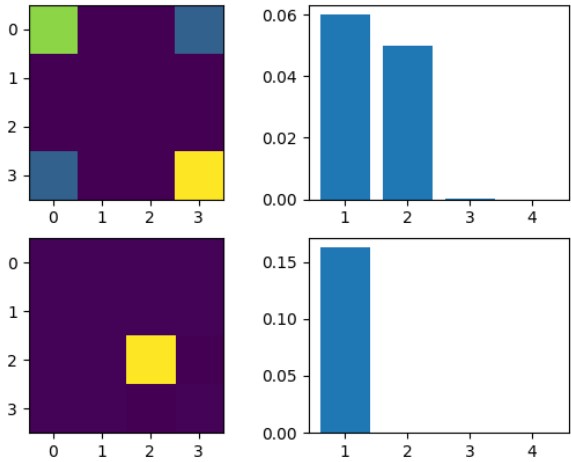

Figure 4: Covariance matrix (left) and spectrum (right) of the hidden layers of a feed forward neural network trained with $L_\Sigma$ regularization to solve the XOR problem. Layer one (top) has learned to utilize unit zero and three while keeping the rest constant, and in layer two only unit two is utilized. This learned structure is the minimal solution to the XOR problem.

### 3.4 NON-LINEAR UNCORRELATED CONVOLUTIONAL FEATURES

Convolutional autoencoders have been used to learn features for visual input and for layer-wise pretraining for image classification tasks. Here, we will see that it is possible to train a deep convolutional autoencoder on real-world data and learn representations that have low covariance, while retaining the reconstruction quality.

To keep it simple, the encoder part of the model used two convolutional layers and two fully connected layers, with a total of roughly 500.000 parameters in the whole model. The regularization

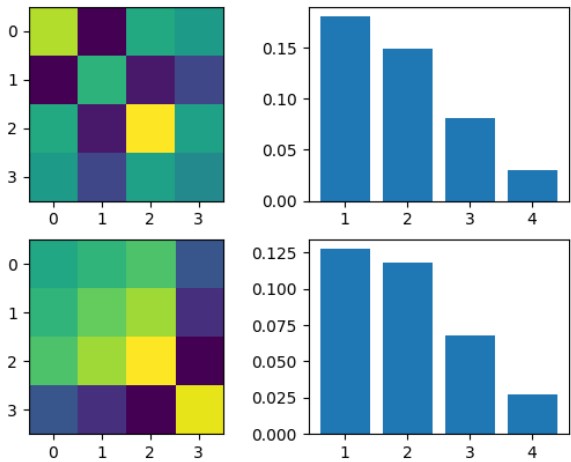

Figure 5: Covariance matrix (left) and spectrum (right) of the hidden layers of a feed forward neural network trained without regularization to solve the XOR problem.

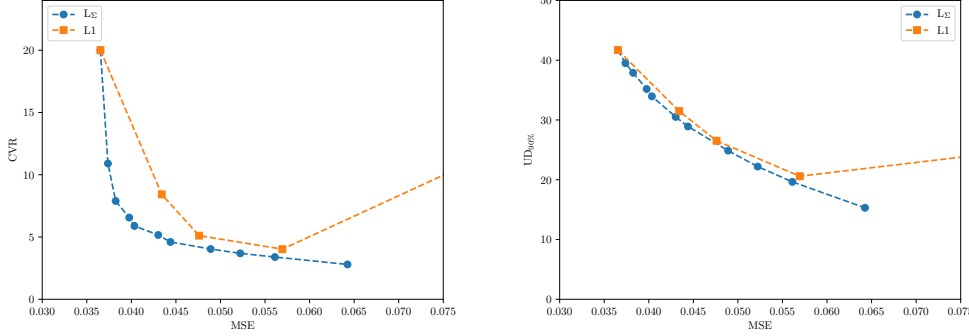

Figure 6: Results from the convolutional autoencoder experiments on CIFAR-10: **Left:** CVR plotted against MSE on the CIFAR-10 test set, using $L_\Sigma$ regularization and L1 regularization, respectively. **Right:** $UD_{90\%}$ plotted against MSE on the CIFAR-10 test set, using $L_\Sigma$ regularization and L1 regularization, respectively. Each point in the plots correspond to doubling the regularization parameter: $\lambda \in [0.0, 0.2, ..., 10.24]$.

was applied to the coding layer which has 84 dimensions, giving a bottleneck effect. The model was trained and evaluated on the CIFAR-10 dataset (Krizhevsky & Hinton, 2009), containing 32x32 pixel colour images tagged with 10 different classes. The model was trained on 45,000 images, while 5,000 were set aside for validation, and 10,000 make out the test set. We compare the results from using $L_\Sigma$ regularization with L1 regularization and with no regularization at all.

The autoencoder was trained with a batch size of 100, using the Adam optimizer (Kingma & Ba, 2015) with an initial learning rate of 0.001. Training was run until the MSE score on the validation set stopped improving[1]. The regularization parameter $\lambda$ was chosen to be 0.08, for a reasonable trade-off between performance and covariance/variance ratio. The reported scores in Table 1 and Figure 6 are averages from training the model five times with different initialization.

---

[1]The source code will be made available when the paper is deanonymized.

The results (see Table 1) show that the high-level features become more disentangled and has a lower CVR (6.56) using $L_\Sigma$ regularization. Without regularization, the score is 20.00, and with L1 regularization the score is 4.03. The model with $L_\Sigma$ regularization obtains a reconstruction error (MSE) of 0.0398, roughly the same as without regularization (0.0365), both of which are much better than using L1 regularization, with an MSE of 0.0569. Figure 6 shows the CVR score plotted against the MSE, illustrating that the $L_\Sigma$ technique leads to more disentangled representations while retaining a better MSE score. As you increase the regularization factor both $L_\Sigma$ regularization pushes down the CVR quickly, while retaining an MSE error that is almost constant. L1 regularization also pushes the model towards learning representation with lower CVR, although slower, and while worsening the MSE error. The $UD_{90\%}$ results show that $L_\Sigma$ encourages representations that concentrate the variation, and the model constantly learns representations with lower $UD_{90\%}$ score than using L1. With $\lambda > 0.08$, the MSE, the $CVR$, and the $UD_{90\%}$ all becomes much worse when using L1 regularization, while the $L_\Sigma$ seems to continue smoothly to improve $CVR$ and $UD_{90\%}$, as the MSE starts to grow.

| Regularizer | CVR | $UD_{90\%}$ | MSE |
|---|---|---|---|
| $L_\Sigma$ | 6.56 | 35.18 | 0.0398 |
| L1 | 4.03 | 20.59 | 0.0569 |
| No regularization | 20.00 | 41.69 | 0.0365 |

Table 1: Results from the convolutional autoencoder experiments on CIFAR. The coding covariance is a normalized sum of covariance over the dimensions of the coding layer. Reproduction MSE is the mean squared error of the reconstructed images produced by the decoder.

## 4    RELATED WORK

Disentanglement is important in learned representations. Different notions of independence have been proposed as useful criteria to learn disentangled representations, and a large body of work has been dedicated to methods that learn such representations.

Principal component analysis (PCA; Pearson, 1901) is a technique that fits a transformation of the (possibly correlated) input into a space of lower dimensionality of linearly uncorrelated variables. Nonlinear extensions of PCA include neural autoencoder models (Kramer, 1991), using a network layout with three hidden layers and with a bottleneck in the middle coding layer, forcing the network to learn a lower-dimensional representation. Self-organizing maps (Kohonen, 1982) and kernel-based models (Schölkopf et al., 1998) have also been proposed for nonlinear PCA.

Independent component analysis (ICA; Hyvärinen et al., 2004) is a set of techniques to learn additive components of the data with a somewhat stronger requirement of statistical independence. A number of approaches have been made on non-linear independent components analysis, (Lappalainen & Honkela, 2000; Honkela & Valpola, 2005). While ICA has a somewhat stronger criterion on the resulting representations, the approaches are generally more involved. Dinh et al., (2015; 2017) proposed a method to train a neural network to transform data into a space with independent components. Using the substitution rule of differentiation as a motivation, they learn bijective transformations, letting them use the neural transformation both to compute the transformed hidden state, to sample from the distribution over the hidden variables, and get a sample in the original data space. The authors used a fixed factorial distribution as prior distribution (i.e. a distribution with independent dimensions), encouraging the model to learn independent representations. The model is demonstrated as a generative model for images, and for inpainting (sampling a part of the image, when the rest of it is given). Achille & Soatto (2017) connected the properties of disentanglement and invariance in neural networks to information theoretic properties. They argue that having invariance to nuisance factors in a network requires that its learned representations to carry minimal information. They propose using the information bottleneck Lagrangian as a regularizer for the weights. Our approach is more flexible and portable, as it can be applied as a regularization to learn uncorrelated components in any gradient-based model that learns internal representations.

Brakel & Bengio (2017) showed that it is possible to adversarial training to make a generative network learn a factorized, independent distribution $p(\mathbf{z})$. The independence criterion (mutual information) makes use of the Kullback-Leibler divergence between the joint distribtion $p(\mathbf{z})$ (represented

by the generator network) and the product of the marginals (which is not explicitly modelled). In this paper, the authors propose to resample from the joint distribution, each time picking only the value for one of the components $z_i$, and let that be the sample from the marginal for that component, $p(z_i)$. A discriminator (the adversary) is simultaneously trained to distinguish the joint from the product of the marginals. One loss function is applied to the output of the discriminator, and one measures the reconstruction error from a decoder reconstructing the input from the joint.

Thomas et al. (2017) considers a reinforcement learning setting where there is an environment with which one can interact during training. The authors trained one policy $\pi_i(a|\mathbf{s})$ for each dimension $i$ of the representation, such that the policy can interact with the environment and learn how to modify the input in a way that modifies the representation only at dimension $i$, without changing any other dimensions. The approach is interesting because it is a setting similar to humans learning by interaction, and this may be an important learning setting for agents in the future, but it is also limited to the setting where you do have the interactive environment, and cannot be applied to other settings discussed above, whereas our approach can.

## 5 CONCLUSIONS

In this paper, we have presented $L_\Sigma$ regularization, a novel regularization scheme based on penalizing the covariance between dimensions of the internal representation learned in a hierarchical model. The proposed regularization scheme helps models learn linearly uncorrelated variables in a non-linear space. While techniques for learning independent components follow criteria that are more strict, our solution is flexible and portable, and can be applied to any feature-learning model that is trained with gradient descent. Our method has no penalty on the performance on tasks evaluated in the experiments, while it does disentangle the data.

We saw that our approach performs well applied to a standard deep convolutional autoencoder on the CIFAR-10 dataset (Krizhevsky & Hinton, 2009); the resulting model performs comparable to the model without $L_\Sigma$ regularization, while we can also see that the covariances between dimensions in the internal representation decrease drastically.

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
