# OpenReview forum: "Disentangled activations in deep networks"
_ICLR.cc/2018/Conference — Reject_

### Official Review · AnonReviewer2 · 2017-11-27
**Simple penalty term enforcing decorrelation in the representation. Seems to work, but not fully analyzed -> so-so manuscript**

**Rating:** 6
**Confidence:** 3

**Review:**

The authors propose a penalization term that enforces decorrelation between the dimensions of the representation
They show that it can be included as additional term in cost functions to train generic models.
The idea is simple and it seems to work for the presented examples.

However, they talk about gradient descent using this extra term, but I'd like to see the derivatives of the
proposed term depending on the parameters of the model (and this depends on the model!). On the other hand,
given the expression of the proposed regulatization,
it seems to lead to non-convex optimization problems which are hard to solve. Any comment on that?.

Moreover, its results are not quantitatively compared to other Non-Linear generalizations of PCA/ICA designed for similar goals (e.g. those cited in the "related work" section or others which have been proved to be consistent non-linear generalizations of PCA such as: Principal Polynomial Analysis, Dimensionality Reduction via Regression that follow the family introduced in the book of Jolliffe, Principal Component Analysis).

Minor points: Fig.1 conveys not that much information.

---

> ### Author Response · Authors · 2018-01-05
> **Comparisons and related work**
>
> Thank you for the comments, we will take them into account. There will be some more evaluations and comparisons added to the final version (see above about Cheung et.al. and Cogswell et.al.).

---

### Official Review · AnonReviewer3 · 2017-11-27
**Benefits are not clear enough**

**Rating:** 5
**Confidence:** 4

**Review:**


I think the first intuition is interesting. However I think the benefits are not clear enough. Maybe finding better examples where the benefits of the proposed regularization are stressed could help.

There is a huge amount of literature about ICA, unmixing, PCA, infomax... based on this principle that go beyond of the proposal. I do not see a clear novelty in the proposal.

For instance the proposed regularization can be achieved by just adding a linear combination at the layer which based on PCA. As shown in [Szegedy et al 2014, "Intriguing properties of neural networks"] adding an extra linear transformation does not change the expressive power of the representation.


- "Inspired by this, we consider a simpler objective: a representation disentangles the data well when its components do not correlate..."

The first paragraph is confusing since jumps from total correlation to correlation without making clear the differences.
Although correlation is a second oder approach to total correlation are not the same. This is extremely important since the whole proposal is based on that.

- Sec 2.1. What prevents the regularization to enforce the weights in the linear layers to be very small and thus minimize the covariance. I think the definition needs to enforce the out-diagonal terms in C to be small with respect to the terms in the diagonal.

- All the evaluation measures are based on linear relations, some of them should take into account non-linear relations (i.e. total correlation, mutual information...) in order to show that the method gets something interesting.

- The first experiment (dim red) is not clear to me. The original dimensionality of the data is 4, and only a linear relation is introduced. I do not understand the dimensionality reduction if the dimensionality of the transformed space is 10. Also the data problem is extremely simple, and it is not clear the didactic benefit of using it. I think a much more complicated data would be more interesting. Besides L_1 is not well defined. If it is L_1 norm on the output coefficients the comparison is misleading.

- Sec 3.3. As in general the model needs to be compared with other regularization techniques to stress its benefits.

- Sec 3.4. Here the comparison makes clear that not a real benefit is obtained with the proposal. The idea behind regularization is to help the model to avoid overfitting and thus improving the quality of the prediction in future samples. However the MSE obtained when not using regularization is the same (or even smaller) than when using it.

---

> ### Author Response · Authors · 2018-01-05
> **Good points!**
>
> Thank you for the insightful comments, we will take them into account when preparing a final version of our paper!

---

### Official Review · AnonReviewer1 · 2017-11-28
**Interesting approach but could use more compelling demonstrations**

**Rating:** 4
**Confidence:** 3

**Review:**

This paper presents a regularization mechanism which penalizes covariance between all dimensions in the latent representation of a neural network. This penalty is meant to disentangle the latent representation by removing shared covariance between each dimension.

While the proposed penalty is described as a novel contribution, there are multiple instances of previous work which use the same type of penalty (Cheung et. al. 2014, Cogswell et. al. 2016). Like this work, Cheung et. al. 2014 propose the XCov penalty which penalizes cross-covariance to disentangle subsets of dimensions in the latent representation of autoencoder models. Cogswell et. al. 2016 also proposes a similar penalty (DeCov) to this work for reducing overfitting in supervised learning.

The novel contribution of the regularizer proposed in this work is that it also penalizes the variance of individual dimensions along with the cross-covariance. Intuitively, this should lead to dimensionality reduction as the model will discard variance in dimensions which are unnecessary for reconstruction. But given the similarity to previous work, the authors need to quantitatively evaluate the value in additionally penalizing variance of each dimension as compared with earlier work. Cogswell et. al. 2016 explicitly remove these terms from their regularizer to prevent the dynamic range of the activations from being unnecessarily rescaled.  It would be helpful to understand how this approach avoids this issues - i.e.,  if you penalize all the variance terms then you could just be arbitrarily rescaling the activities, so what prevents this trivial solution?

There doesn't appear to be a definition of the L1 penalty this paper compares against and it's unclear why this is a reasonable baseline. The evaluation metrics this work uses (MAPC, CVR, TdV, UD) need to be justified more in the absence of their use in previous work. While they evaluate their method on non-toy dataset such as CIFAR, they do not show what the actual utility of their proposed regularizer serves for such a dataset beyond having no-regularization at all. Again, the utility of the evaluation metrics proposed in this work is unclear.

The toy examples are kind of interesting but it would be more compelling if the dimensionality reduction aspect extended to real datasets.

> Our method has no penalty on the performance on tasks evaluated in the experiments, while it does disentangle the data

This needs to be expanded in the results as all the results presented appear to show Mean Squared Error increasing when increasing the weight of the regularization penalty.

---

> ### Author Response · Authors · 2018-01-05
> **Clarification on related work**
>
> Thank you for constructive and well-researched comments. Please consider the following comments on the mentioned related work.
>
> Cheung, et.al.  - This paper describes a different type of regularizer that penalize the correlation between hidden units and labels. In contrast we aim to learn a hidden representation that disentangles unknown underlying factors by penalizing correlation between hidden units. Hence, our method use no labels and can go much further in disentangling the signal.
>
> Cogswell et.al. - As noted by reviewer one, what sets our work apart from Cogswell et.al., is that we penalize the full covariance matrix while they only penalize the off diagonal elements. The reason that we included the diagonal is that this will lead to a lower variance hypothesis, by removing information not needed to solve the task from the representation, which in turn yields guaranteed lower excess risk (Maurer 2009). Also, we got slightly better empirical results doing this and will include comparisons to Cogswell et.al. in the final version of the paper.
> Another difference between our model and that of Cogswell et.al. is that we use the L1 matrix norm (in contrast to the frobenius norm) in order to promote a sparse solution.

---

### Decision · Program_Chairs · 2018-01-29
**ICLR 2018 Conference Acceptance Decision**

**Decision:**

Reject

**Comment:**

The novelty of the paper is limited and it lacks on comparisons with relevant baselines, as pointed out by the reviewers.